# Towards a Deep Network Architecture for Structured Smoothness

**Haroun Habeeb**
Department of Computer Science
University of Illinois at Urbana Champaign
Champaign, IL 61820
`haroun7@gmail.com`

**Sanmi Koyejo**
Department of Computer Science
University of Illinois at Urbana Champaign
Champaign, IL 61820
`sanmi@illinois.edu`

## Abstract

We propose the Fixed Grouping Layer (FGL); a novel feedforward layer designed to incorporate the inductive bias of structured smoothness into a deep learning model. FGL achieves this goal by connecting nodes across layers based on spatial similarity. The use of structured smoothness, as implemented by FGL, is motivated by applications to structured spatial data, which is, in turn, motivated by domain knowledge. The proposed model architecture outperforms conventional neural network architectures across a variety of simulated and real datasets with structured smoothness.

## 1 Introduction

The effectiveness of predictive models often depends on the choice of inductive bias, and the extent to which this inductive bias captures real-world structure. For instance, one example of such bias encoding leading to improved performance is convolution. In principle, convolutional weights could be learned directly from data. However, in practice, imposing this structure leads to improved performance when compared to fully connected models, and as a result, Convolutional neural networks (CNNs) have enjoyed wide use for computer vision tasks (Krizhevsky et al., 2012). Similarly, recurrent neural networks such as LSTMs are effective for text (Sundermeyer et al., 2012), and certain graphical models are ideal for sentence segmentation and labeling (Lafferty et al., 2001). Our work follows this philosophy. Specifically, we propose a feedforward layer for deep neural networks that is suitable for neuroimaging and potentially useful for other data where variables can be grouped due to underlying structure.

Data with multiple input variables often exhibit some structure. For example, the El Nino dataset (Bay et al., 2000) consists of measurements by weather buoys in the ocean, and one expects that nearby buoys can be grouped together. Similarly, socio-economic data can often be grouped together by geographic proximity. Financial market data of individual stocks can be grouped together based on the industrial sector to which a company belongs. Along similar lines, brain parcellations are a well studied paradigm for capturing the structure of brain activity (Thirion et al., 2014), often via statistical parcellation based on ward clustering (Ward Jr, 1963). The result of ward clustering is a tree where leaf nodes represent voxels of the brain and interior nodes represent grouping of voxels into spatial clusters. Figure 1 visualizes the output of ward clustering at various granularities when applied to the human connectome project resting state brain data (Van Essen et al., 2012).

*Contributions:* Our primary technical contribution is the **Fixed Grouping Layer** (**FGL**). FGL is designed to extract features within each group, and additionally guarantees that each output vector is only affected by the input vectors related to it by the specified grouping. We demonstrate the benefit of using FGL on simulated experiments and real neuroimaging data. We compare FGL against fully connected networks, convolutional neural networks, CoordConv (Liu et al., 2018), and a closely related method proposed by Aydore et al. (2018). We extensively evaluate the performance of FGL on simulated and real brain imaging data showing improved performance.

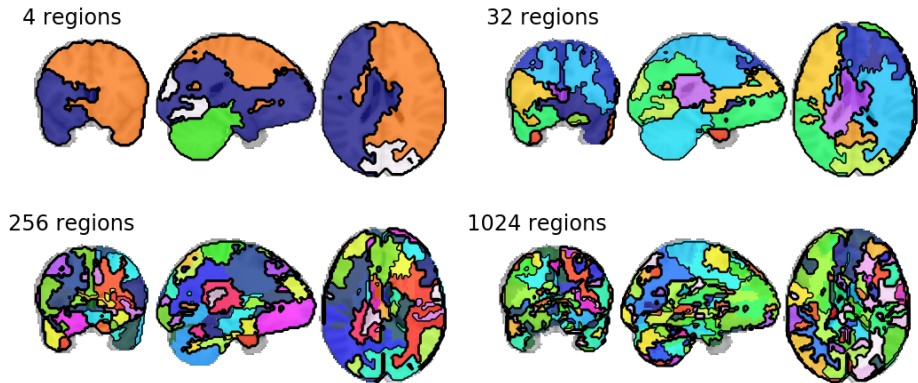

Figure 1: Computed brain parcellation at various granularities (details in the text). The text in the top left indicates the number of regions. Each color in each figure corresponds to a region/group. Notice the consistency between parcellations while increasing granularity.

## 1.1 DECODING BRAIN IMAGES: BACKGROUND AND RELATED WORK

Functional Magnetic Resonance Imaging (fMRI) is a popular brain imaging technique which measures a physiological correlate of neuron activity (Huettel et al., 2004). The brain imaging scans are generally of two kinds: resting state and task data. Resting state data (rfMRI) is collected while the subject is at rest, i.e., while the subject is not actively engaged in a task. Task data (tfMRI) is collected while the subject is engaged in a predefined task, for example, a motor task such as moving their fingers. fMRI data can be represented as 3-dimensional images, and have rich structure that has been studied extensively, including in the ML literature (Koyejo et al., 2014; Park et al., 2013). Importantly, coarse correspondences have been discovered between brain regions and specific functions or behavior (Frost and Goebel, 2012; Sporns, 2013)

We particularly focus on brain decoding – a standard task in fMRI brain data analysis where the brain image is used to predict the associated task or stimulus. Broadly, there are two types of brain decoding models: end to end models and models which perform dimensionality reduction followed by a low-dimensional prediction. On one hand, dimension reduction directly captures the notion of grouping variables together. On the other hand, end to end models rarely employ brain spatial structure. This observation motivates our work. In recent years, decoding from fMRI studies has been attempted using a variety of methods: Bzdok et al. (2015) use factored logistic regression while Sarraf and Tofighi (2016) use convolutional neural networks. Mensch et al. (2017) use a factored model after performing dimensionality reduction. Inspired by similar motivations, Aydore et al. (2018) construct a regularizer using feature groupings obtained from a fast clustering method. They demonstrate that such a regularizer outperforms dropout and $L_2$ regularization. However, they do not consider employing this structure in a deep model, opting instead for a wide shallow approach. Compared to Aydore et al. (2018), our results illustrate the benefits of depth combined with spatial sparsity for brain image decoding.

## 2 FIXED GROUPING LAYER

Next, we formally define our idea of groups. Given a set of input variables $\mathcal{X} = \{x_i : 0 \leq i < n_{in}, i \in \mathbb{Z}\}$, a grouping of variables, denoted by $\mathcal{G}$ is a subset of the power-set of $\mathcal{X}$ such that each $x_i$ is in at least one set in $\mathcal{G}$. That is,

$$\mathcal{G} \subset \mathbf{2}^{\mathcal{X}}, \text{ such that, } \forall x_i \in \mathcal{X} : \ x_i \in \bigcup \mathcal{G}.$$

Each set in $\mathcal{G}$ is a group of variables. For example, in the case of a colored image, each pixel can be considered a variable with an associated feature vector of length 3, i.e., each $x_i$ represents a pixel and $x_i \in [0, 1]^3$. A spatially smooth grouping of these variables corresponds to a segmentation of the image. Optionally, the groups can be mutually exclusive.

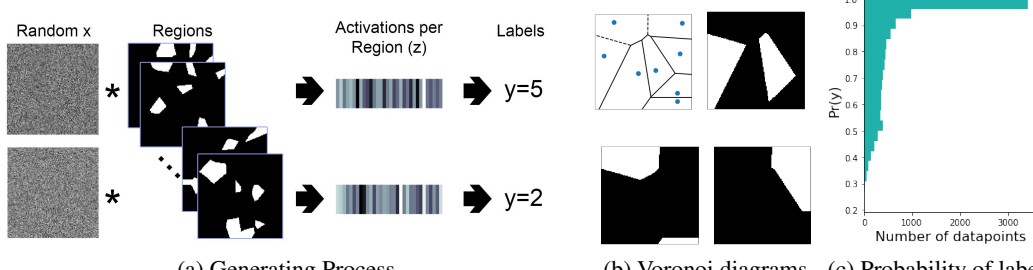

(a) Generating Process        (b) Voronoi diagrams    (c) Probability of label

Figure 2: **(a) Simulated Dataset**: Random images are aggregated over regions/groupings to obtain activations ($z$) that are used to assign labels. Groupings are comprised of multiple smaller regions, all spatially connected. **(b) Voronoi diagrams and induced groups**. The upper left image is a Voronoi diagram of 8 random sites (blue points) and lines that partition space into sets of points closest to the same site. The other images are possible groupings using 2 partitions. **(c) Histogram of probability of assigned label**. The y-axis is probability and x-axis is number of points. Most datapoints have $Pr(y) > 0.5$ (labels are low-noise).

An FGL layer takes as input $n_{in}$ vectors of $c_{in}$ length each, where the $n_{in}$ vectors are grouped into $n_{out}$ groups. Further, let $c_{out}$ be the length of the vector associated with each group. Note that these groups do not have to be mutually exclusive, but mutually exclusive groups offer benefits that we describe in the supplementary. The model architecture is allowed to use multiple channels (analogous to the # of filters of standard convolutional networks). $c_{in}$ and $c_{out}$ are the number of input and output channels. Mathematically, the Fixed Grouping Layer is given by:

$$z = A((xv) \odot u) + b,$$

where: $z \in \mathbb{R}^{n_{out}, c_{out}}$ is the matrix representing the output with each row represents one group. $x \in \mathbb{R}^{n_{in}, c_{in}}$ is a matrix representing the input - one row for each input vector. $A$ is a binary matrix that represents the *grouping* - $A_{j,i} = 1$ if and only if $x_i$ is in group $j$. $\odot$ represents the Hadamard product (elementwise multiplication) (Horn, 1990). $u, v, b$ are parameters of the model. $v$ is used for a linear transformation from $\mathbb{R}^{c_{in}}$ to $\mathbb{R}^{c_{out}}$, i.e., $v \in \mathbb{R}^{c_{in}, c_{out}}$. $u$ is a matrix of size $n_{in} \times c_{out}$. $b$ the bias, is a matrix of size $n_{out} \times c_{out}$ We denote the $i^{th}$ input vector, i.e., the $i^{th}$ row of $x$, by $x_i$. Observe that FGL is a fully connected layer when there is only one group which contains all variables.

## 3 EXPERIMENTS

We construct a deep neural network for classification using repeated layers of FGL (and activation functions), followed by either a fully connected network or an FGL that groups all inputs into a single group. This is inspired from traditional CNN based classification models. We provide a visualization of a simplified model in Figure S2.

**Regularization**: We use weight normalization (Salimans and Kingma, 2016) , which is a reparameterization of the weights of a neural network that decouples the norm and the direction of the weights. That is, for a single dimension of a fully connected layer , weight normalization reparameterizes $w$ as $w = g(\theta/||\theta||)$, where $\theta$ is a vector of the same length as $w$ and $g$ is a scalar. The network now learns $g, \theta$ instead of $w$. For FGL we apply weight norm on both $u$ and $v$. Weight norm is applied by treating $u$ as $c_{out}$ different vectors and $v$ as the weights of a fully connected layer.

We use `Pytorch` (Paszke et al., 2017) to implement models, and `nilearn` (Abraham et al., 2014) to preprocess and visualize fMRI images. Training was done using Adam (Kingma and Ba, 2014) on 4 K80 GPUs. Code is available at `https://www.github.com/anon/repo` and a minimal version is provided in the supplementary.

### 3.1 BASELINES

We consider brain decoding as a classification task, and use two common types of models as baselines: fully connected networks and convolution based models such as standard Convolutional

Neural Networks (CNNs) and their CoordConv variant. To the best of our knowledge, our baselines include state of the art approaches on the considered datasets.

**Multinomial Logistic Regression (LR)**: Multinomial logistic regression is a standard model (Bishop, 2006a), given by: $\hat{y} = softmax(Wx + b)$, for an input $x \in \mathbb{R}^d$, parameterized by weights $W \in \mathbb{R}^{k,d}$ and bias $b \in \mathbb{R}^k$ where $k$ is the number of possible labels. Here, $\hat{y}$ is the vector of predicted probabilities for each class or label. Clearly multinomial logistic regression by itself uses no spatial information.

**Feedforward Neural Networks (FNN)**: FNNs are ideal for tasks where the input has neither a grid-like structure nor a sequential structure. The architecture is an alternating sequence of linear transformations and activation functions (Goodfellow et al., 2016).

**Convolutional Neural Networks (CNNs)**: CNNs LeCun et al. (1995), are a popular tool in deep learning. Many problems, where the inputs have a spatial representation, lend themselves to convolution. CNNs are popular not only for their flexibility but also because of the assumptions they make about the nature of the input - one of them being that dependencies between pixels are local and spatially invariant. These assumptions are usually appropriate for natural images. However, in the case of brain fMRI, since features are also dependent on position, i.e., features are not position invariant, CNNs might not work as well.

**CoordConv (CC)**: Liu et al. (2018) demonstrate that CNNs are sometimes unable to capture location-dependent spatial representations. e.g. transforming a pair of coordinates to a one-hot representation. One reason for this failure could be that the coordinate transformation problem directly conflicts with the underlying assumptions of CNNs. They propose the CoordConv layer as a solution. CoordConv is essentially a convolutional layer except that it includes coordinates as additional input channels. Thus, CoordConv enables CNNs where local dependencies change across the image.

## 3.2 SIMULATED DATA

First, we discuss grouping via *Voronoi Diagrams*. Given a set of points, called sites, a Voronoi diagram (Aurenhammer, 1991; Okabe et al., 2009) is the division of a plane into regions based on distance to the sites. Usually, positions whose closest site is the same are grouped together. Consider the grouping induced by a set of $m$ sites $P = \{p_i : p_i \in [0, s]^2, 0 \le i < m\}$ for some $s$ indicating the size of the plane:

$$g_i = \{x_j : \min_k |x_j - p_k| = i\} \, \forall \, 0 \le i < m.$$

We use Voronoi diagrams because they create regions which are spatially connected. To increase the complexity of the task, we use groupings which are unions of arbitrarily chosen Voronoi regions - resulting in groups comprised of multiple spatially connected regions which may not be connected to each other. We provide an example in Figure 2b.

Consider input data sampled from a Gaussian prior: $x \sim \mathcal{N}(\mathbf{0}, S)$, where $\mathbf{0}$ is a zero-vector and $S$ is an arbitrary covariance matrix of appropriate size. We let $x \in \mathbb{R}^{s^2}$ for an integer $s$ - the idea being that $x$ is a flattened version of a grayscale image of size $s \times s$. Next, suppose that datapoints are labelled based on a linear function. That is, $z|x \sim \mathcal{N}(Fx, \Sigma)$, for a fixed covariance matrix $\Sigma$, and a matrix $F$ of size $k \times s^2$ where $k$ is the number of labels. The label, $y$, is assigned based on $z$: $y = \arg\max_i softmax(Wz)_i$. for a full rank matrix $W$.

We briefly analyze this simple generative model in the context of FGL. Using conjugate priors (Bishop, 2006b), it is straightforward to show that,

$$x|z \sim \mathcal{N}(F^\top \Sigma^{-1} z, (S^{-1} + F^\top \Sigma^{-1} F)^{-1}). \tag{1}$$

To explore the implications of equation (1), consider an $F$ that is sparse such that the non-zero positions in each row of $F$ correspond to a segmentation of the input image (a grouping of pixels). For example, circular patches on the image, or in our case, Voronoi diagrams. Additionally, if $\Sigma$ is an identity matrix, then each dimension of $z$ corresponds to the sum of values of a group of pixels.

**Dataset**: We create a dataset which samples $x$ from the the Gaussian prior with $S$ being an identity matrix. We use $s = 128$ so that each $x$ can be interpreted as a square image. We create $F$ by first creating a Voronoi diagram of 512 randomly selected points, and then merging these regions into $k = 32$ groups. We sample $z$ from the the corresponding conditional distribution. We fix a random $W$ and then assign the label $y$ with highest likelihood to the datapoint $x$. We sample 50000 points to

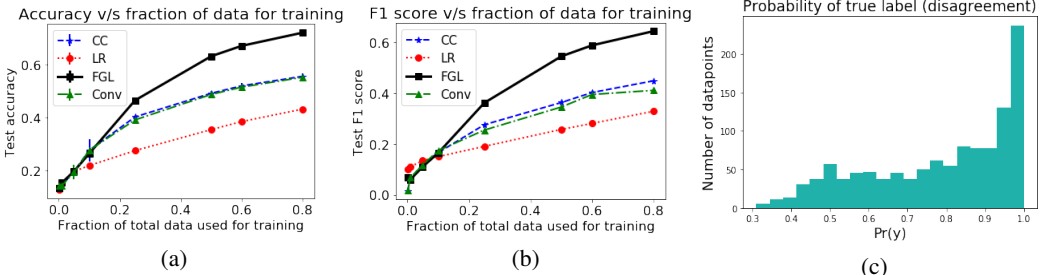

Figure 3: **(a)** Test accuracy (with error bars) on held out 20% of simulated dataset vs. fraction of data used for training. The graph indicates that FGL has better empirical sample complexity. The small magnitude of error in estimation of performance indicates that models are well trained and the difference is due to the models themselves. **(b)** Minimum (across classes) F1 Score on held out test set vs. fraction of data used for training. The difference in performance is not due to performance on a single class/region, but rather across all labels. **(c)** Histogram of ground truth probability of labels for points where FGL is correct but CNN misclassifies. This demonstrates that the misclassification by CNN is not only on noisy datapoints but also for datapoints where the label should be clear.

create the simulated dataset. A visualization of the process is provided in Figure 2a. To ensure that the dataset wasn't too noisy, we plot a histogram of probability of assigned label in Figure 2c. The histogram shows that only a small number of datapoints are noisy - in most cases, the assigned label has a probability of at least $0.5$.

### 3.2.1 MODELS

To demonstrate the benefit of using the Voronoi Diagram during classification, we train 4 models - Logistic Regression (**LR**), a Convolutional Neural Network (**Conv**), a CoordConv variant (**CC**) of the same CNN, and a model using our proposed layer - FGL followed by a fully connected network. Our FGL model is provided the voronoi regions. The number of parameters in each model is roughly the same. Since the dataset uses labels that are linear in terms of $x$, we use no non-linear activations in any of our models. We found that using maxpooling in the CNN and CoordConv hurt performance. We don't report results with an FNN because it performs similar to LR.

### 3.2.2 PROCEDURE AND ANALYSIS

We create a test set using 20% of the simulated dataset. The remaining points are used for training. For each model, we train using various quantities of available data, and test on the held out set. The results are aggregated over 10 runs – with a randomly sampled test set for each run. A plot of the test accuracy vs. fraction of data used for training is given in Figure 3. We find that the standard deviation of accuracies of these models is small - indicating that the failures are not due to poor initialization or poor training but rather a difference in models.

This experiment was designed to demonstrate a failure of convolution based models and also fully connected methods. Although this satisfies our intuition that using spatial structure should help drastically improve performance, we investigate the datapoints at which the CNN failed but FGL did not. The first thing to check was the probability of assigned labels for these points - a histogram of the same for a random subset of the testing set is provided in Figure 3c. The next sanity check is to ensure that the drop in performance isn't just for one set of regions or one class. To that end, we plot the lowest F1 score (lowest across classes) in Figure 3b. We see the same trend - FGL performs better than CNNs, CoordConv and Logistic Regression. These plots indicate the validity of the gain in performance - it is due to neither noisy labels, nor failure on any one label. Hence, using a grouping of variables seems to provide a significant benefit.

### 3.3 fMRI CONSTRAST PREDICTION

We evaluate our models on 5 datasets which were used by Mensch et al. (2017): **Archi** (Pinel et al., 2007), **Brainomics** (Orfanos et al., 2017), **Cam-CAN** (Shafto et al., 2014), **LA5c** (Poldrack et al.,

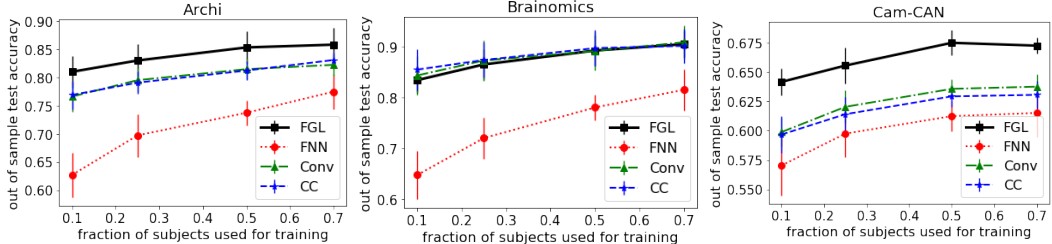

Figure 4: **Test accuracy**: Out of sample accuracy measured on 30% of dataset v/s fraction of subjects used for training. FGL performs well even when a small amount of data is used for training.

Table 1: Test Accuracy per dataset, model. P-values are dataset-specific – in this case we use the Wilcoxon test to compare our method to each of the others on the HCP dataset. We reported p-values for HCP only, as the dataset was large, and the performance improvement gaps were not extremely obvious. *p*-value from one sided Wilcoxon rank sum test shows that FGL (3 layer) is better than other models on HCP dataset.

|  | Archi | Br | Cam | HCP | LA5c | *p*-value (HCP) |
|---|---|---|---|---|---|---|
| LR | 81.00% | 74.42% | 63.29% | 91.70% | 61.12% | 5.413e-06 |
| FNN | 82.72% | 81.47% | 61.52% | 92.16% | 60.86% | 3.789e-05 |
| Conv | 84.23% | **90.85%** | 63.77% | 91.38% | 61.99% | 5.413e-06 |
| CC | 83.96% | 90.64% | 63.07% | 91.52% | 62.04% | 9.083e-05 |
| FGL (1 Layer) | 85.78% | 88.65% | 67.23% | 92.70% | 64.49% | 0.001097 |
| FGL (2 Layers) | 85.78% | 89.87% | **67.46%** | 92.67% | **64.57%** | 0.000525 |
| FGL (3 Layers) | **87.07%** | 90.38% | 67.27% | **93.36%** | 64.24% | – |
| Feature Grouping (b=10) | 75.80% | - | 59.00% | - | 53.09% | – |
| Feature Grouping (b=50) | 76.81% | - | 59.77% | - | 57.17% | – |
| Feature Grouping (b=100) | 78.55% | - | 59.66% | - | 60.54% | – |
| Feature Grouping (b=200) | 73.48% | - | 58.56% | - | 53.79% | – |

2016) and **HCP** (Van Essen et al., 2012). They have 78, 94, 605, 191 and 787 subjects respectively. Archi, Brainomics, Cam-CAN and LA5c are small datasets with 2340, 1786, 3025 and 5756 images respectively. On the other hand, HCP is larger with 18070 images. These datasets have different sets of labels corresponding to different cognitive processes. Each dataset is publicly available on `NeuroVault`[1] (Gorgolewski et al., 2015), an aggregation of fMRI datasets. Our only required preprocessing for the task contrast data was to upsample Cam-CAN and Brainomics to the same resolution as HCP.

The assessment of fMRI decoding is an important topic in its own right due to unique spatial characteristics and typical sample sizes. Varoquaux et al. (2017) show that leave-one-out strategies for cross validation can be unstable, and suggest that using reasonable defaults is a good strategy. Additionally, it is well known that having common subjects between train and test datasets can lead to misleading results. This is because such a test set does not measure how well a model can generalize from one subject to another. Hence, we evaluate models via out-of-sample accuracy, i.e., we hold out some subjects (30%) for the testing dataset in each run. Further, we train all models with reasonable defaults that we found were not critical for model performance.

### 3.3.1 PARCELLATION

Since Thirion et al. (2014) showed that ward clustering provides good parcellations of the brain, we perform ward clustering on a fraction of HCP resting state data (total size of 4TB). We downsample the resting state time series data to about 1% of the original frequency, and use the brain activations at each position as the feature vectors for clustering. The downsampling is needed due to hardware constraints. We did not use task datasets for clustering since we would have to hold out more data for clustering - exacerbating data scarcity. Additionally, resting state data is more easily acquired

---

[1] `https://neurovault.org/`

(Biswal et al., 2010), and there are strong correlations between tfMRI and rfMRI (Smith et al., 2009). Thus, using rfMRI should provide a good if not better parcellation of the brain.

To make a deep network using FGL we require a hierarchical clustering and not just a parcellation. Hence, instead of using the segmentation produced by the parcellation algorithm provided by `nilearn`, we use the computed ward tree. We then slice into the ward clustering to produce parcellations with 32, 256 and 1024 regions. These have been visualized in Figure 1. Clearly, these groups are spatially connected. We need groupings of voxels into 1024 groups, then a grouping of these 1024 groups into 256 groups and finally a grouping of 256 groups into 32 groups. Since ward clustering is a hierarchical clustering scheme and outputs a tree structure, we extract these groupings by making appropriate cuts into the tree.

### 3.3.2 Models

**Fully Connected Models**: We experimented with Fully Connected Neural Networks (FNN) and Multinomial Logistic Regression (LR) with and without Dimension Reduction using our parcellation. We found that using Dimension Reduction reduced performance and hence do not report it. For FNNs, we tried 2 and 3 layer versions with intermediate sizes chosen from 64, 128, 256 and 512. The model with intermediate layer sizes of 512 and 128 worked best. The aforementioned models take a masked fMRI image as input and we used the MNI152 mask provided by `nilearn`. Emperically, we find that, for brain decoding, using a linear activation performs better than using non-linear activations. For this reason, we use a linear activation for our models. Unfortunately, we are unclear about why this occurs for this domain. We also evaluate *Feature Grouping* suggested by Aydore et al. (2018) which is also designed to exploit structured sparsity but does this using a wide model (implemented using multiple randomized clusterings), unlike our deep FGL approach. We used code provided by the authors.

**Convolutional Neural Networks (Conv, CC)**: We experimented with a variety of architectures and found no improvement by using residual connections or Batch-Norm. We also report results using CoordConv. We found that using non-linear activations hurt the model's performance, similar to our finding with FNNs. Further, maxpooling also reduced performance. The architecture is 5 3-D convolution layers of stride 2 and kernel size 4. The input volumes have size $91 \times 109 \times 91$, and convolution reduces the volume to $2 \times 3 \times 2$ with 128 channels. We flatten this volume and pass it through a fully connected network to get the score for each label. The architecture for the CoordConv is identical to the CNN since CoordConv only concatenates a few input channels to the input image. We use **Conv** to refer to the Convolutional Neural Network and **CC** to refer to the CoordConv variant.

**FGL**: We use 3 layers of FGL, each of which use the Parcellation described earlier. The input images have 212455 voxels after masking. We treat every voxel as a variable with a single feature. These voxels are then reduced to 1024 groups with feature vectors of length 8 each. Next, these groups are reduced to 256 variables with 64 features and finally to 32 variables with 128 features. The final prediction is made by flattening the output of the last FGL layer and passing it through a fully connected layer. The resulting number of parameters is roughly 2 million, which is also roughly the same number of parameters used for CNN and CC. While this is a lot of parameters, we found that reducing the number of parameters by changing the number of features for each intermediate variable decreases performance for both convolution and FGL.

### 3.3.3 Procedure and results

We split each dataset multiple (10) times into a train and test set. The split is done such that no subject in the test set appears in the training set. In each split, 30% of subjects are used for testing, and all or a part of the remaining subjects are used for training. Convolution based models were trained for 50 epochs, feedforward neural networks for 30 and FGL for 20. These hyperparameters were selected by monitoring for overfitting on the training set (using a further validation split). We perform experiments to study: (1) the benefit of FGL given a lot of data, (2) the benefits of FGL at small sample sizes, (3) the effect of depth, and (4) the effect of intermediate vector length.

**Large sample setting**: The first experiment uses all of the training data (70% of total data) to train. We report out-of-sample accuracy in Table 1. We also report the *p*-values of one sided Wilcoxon rank sum tests between the performance of each model compared to 3 Layer FGL on the HCP dataset.

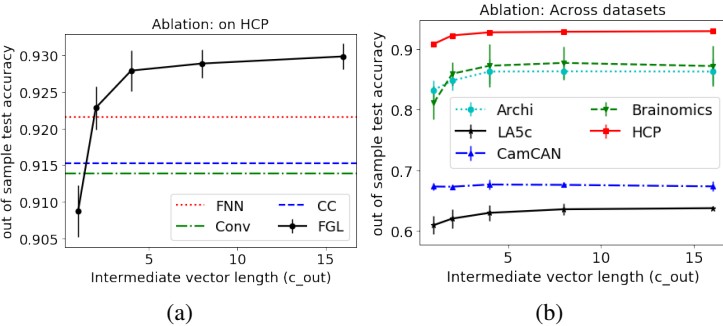

Figure 5: **Ablation of intermediate vector length**: (a) On HCP, along with baselines. (b) On all datasets. Increasing the intermediate vector length improves performance, except on Cam-CAN.

**Small sample setting**: The second set of experiments varies the fraction of data used for training on the smaller datasets - namely, Archi, Brainomics and Cam-CAN. We explore small sample performance because limited sample sizes are typical for fMRI studies. Figure 4 plots test accuracy against fraction of data used for training. It demonstrates that FGL outperforms the baselines even when only a small amount of training data is used.

**Effect of depth, FGL depth vs. width**: We train two additional models to study the effect of depth: one which uses only the first layer of FGL, and another that uses the first two layers. In Table 1, they're referred to as "FGL (1 Layer)" and "FGL (2 Layers)" respectively. The results show that increasing FGL depth provides a statistically significant improvement in test accuracy on the HCP dataset. When compared to Feature grouping (Aydore et al., 2018), the results in Table 1 show that even after extensive tuning (we used the reported best performing settings and attempted additional tuning), this approach is not competitive – suggesting a significant representation benefit by exploiting FGL depth vs. width for this task. Unfortunately, the provided code did not scale to the largest datasets – hence the missing results on HCP.

**Effect of intermediate vector length**: To study the effect of intermediate vector length, we train five single layer FGL models with intermediate vector lengths ($c_{out}$) of 1, 2, 4, 8 and 16. We plot the test accuracy against $c_{out}$ in Figures 5a and 5b. On 4 out of 5 datasets, increasing $c_{out}$ improves test accuracy. This is expected because the classification task is not binary and each region of the brain could contribute to multiple cognitive processes in different ways. However, the effect is not as pronounced on the Cam-CAN dataset. We suspect that this is because Cam-CAN has fewer labels (5) than the other datasets. We note that $c_{out}$ could be interpreted as width of our model, similar to channel size of CNNs, which is known to be important.

These experiments demonstrate the clear benefit of using FGL compared to other models with roughly the same number of parameters. When using 70% of data for training, FGL provides 2-6% of improvement in test accuracy on 4 of 5 datasets. A similar trend exists even when smaller amounts of data is used. As for the 5th dataset, Brainomics, FGL is on par with CNN based methods but better than Fully connected networks. While the effect of depth is not clear on smaller datasets, we note that on the HCP dataset, deeper models have a statistically significant improvement in performance. Further, an increase in $c_{out}$ also improves performance.

Our main argument is that current methods discard important information about spatial smoothness as encoded by hierarchical spatial clustering. As pointed out, the main cost of our method is in constructing these hierarchies. For brain imaging, most datasets include readily available resting state data. From the larger view, we plan to encourage application communities to develop application-specific FGL architectures, which can be shared across several related tasks.

## 4 CONCLUSION

In this work we propose a new layer architecture, the Fixed Grouping Layer (FGL), parameterized by a grouping of input variables. FGL explicitly extracts features within each input group. This is in

contrast to convolution which extracts local features across the input, and fully connected networks which extract both global and local features. We demonstrate the benefit of using FGL on 5 real fMRI datasets of different sizes. Future work will involve the application of FGL to other tasks and application domains.

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

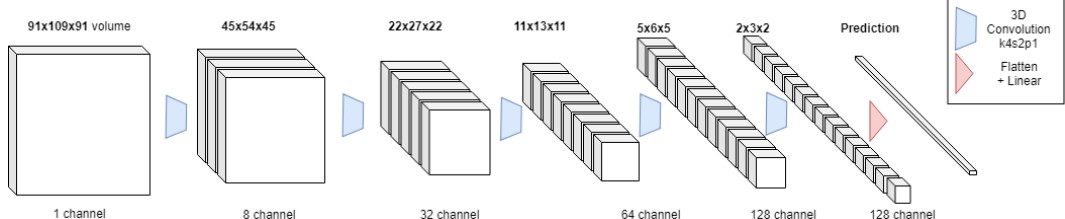

Figure S1: **Baseline**: The architecture used for Convolutional Neural Networks and CoordConv. In the CoordConv variant, each volume is concantenated with 3 channels for coordinate along each dimension being passed in as input. The volumes are written above each volume while the number of channels for each volume is written underneath it.

Table S1: Test Accuracy to study imperfect A: Reported numbers for each model are averaged over multiple (3) runs.

| Model | Test accuracy |
|---|---|
| LR | 43.0% |
| Conv | 56.4% |
| CC | 56.6%% |
| FGL (16 clusters) | 61.3% |
| FGL (32 clusters) | 66.1% |
| FGL (48 clusters) | 66.3% |
| FGL (perfect $A$) | 70.2% |

# Supplementary Materials

## S1  FGL DETAILS

**Input Specification**: In this work we deal with image-like data, either in 2D or 3D. Consider an input with $s$ pixels or voxels in $c$ channels - for example, a $64 \times 64$ image with RGB colors will have $s = 4096$ and $c = 3$. Such an input is treated as $s$ variables with feature vectors of $c$ length for each variable.

**Groupings**: Since the output of the first FGL layer is feature vectors for each group, the grouping of the second FGL layer must group together the outputs of the first layer. Hence, we need a hierarchical structure with input variables at the leaf nodes. In this work, we use a Ward clustering of the brain, however, other clusterings may be more appropriate in other settings.

## S2  ABLATION: IMPERFECT A

We ran an experiment using the simulated dataset to estimate how robust FGL was to imperfect $A$. Apart from providing FGL the true voronoi diagrams, we also ran FGL using the clusters from K-Means clustering That is, $A_{ji} = 1$ if pixel $i$ was in cluster $j$ according to K-Means. We did this using 16, 32 and 48 clusters obtained by clustering pixels in the training dataset. The results are reported in S1. We see that although there is a drop when using imperfect $A$, FGL still outperforms logistic regression and convolution. This emphasises the benefit of capturing structure.

We do see a larger drop when we use a clustering that is not representative enough (16 clusters), but don't see much gain from using a more representative one (48 clusters).

## S3  FGL VARIANTS

While the FGL model is straightforward, multiple variants of it are possible. First, notice that FGL is essentially the following three operations (ignoring the bias):

- **Linear transformation**: The multiplication, $xv$, transforms the data $x$ from one basis to another using a linear transform $v$.
- **Rescaling**: The hadamard product with $u$ rescales each vector along each dimension independently
- **Aggregation**: The multiplication by $A$ aggregates the vectors $(xv) \odot u$ within each group using summation.

Performing these operations in a different order creates some basic variants: for example, we could aggregate within groups, then rescale, and finally perform a linear transformation. These changes to operation order will require the parameters to be defined differently. For example, if the hadamard product with $u$ is done after aggregation, then $u$ will need to have $n_{out}$ rows.

## S3.1  REDUCTIONS

Another interesting variant is to replace the aggregation with a `max` operation within each group along each dimension. We think this is similar to doing a `maxpool` operation in convolution neural networks while the summation by $A$ is similar to a weighted-sum-pool depending on the values of $A_{ji}$. Early results showed worse results when using the `max` reduction variant and hence we did not investigate it further. However, it might prove effective in cases where a signal being present in one variable within a group is equivalent to the group showing that variable.

## S3.2  MULTIPLE GROUPINGS

Another possible benefit that we do not investigate is the use of multiple variable groupings - we can concatenate the $A$ matrices that represent each grouping to make FGL extract features within each group from the union of both groups. That is, if $A^{(0)}, A^{(1)}$ are the matrices that represent two groupings, we could use $A = [A^{(0)}\ A^{(1)}]$. This would allow one to make use of multiple types of groupings. For example, we could create parcellations at different points of the accuracy-reproducibility tradeoff studied by Thirion et al. (2014), and make use of both. Similarly, one could create parcellations from different datasets and use them at once. However, using a single parcellation was sufficient to create a significant gain in performance, hence we don't go deeper in this direction. We mention a few other variants that did not perform as well in supplement S3.

# S4  IMPLEMENTATION

Our implementation of FGL using `PyTorch` is available at `https://github.com/anonymous/link`. In this section, we discuss some challenges in implementation - If the number of input variables is large, performing $A((xv) \odot u)$ as a matrix multiplication is expensive. There are some ways to work around this:

- Since $A$ is a binary matrix, we can treat $((xv) \odot u)$ as a matrix of embeddings, and lookup the indices at which $A$ is non-zero, then performing necessary aggregation.
- If the variable groups are mutually exclusive - that is, each input variable only belongs to one group, then $A((xv) \odot u)$ can be performed by `scattering` $(xv) \odot u$ according to the indices at which $A$ is non zero.

## S4.1  FGL INITIALIZATION

Prior literature (He et al., 2015; Glorot and Bengio, 2010; Sutskever et al., 2013) has shown that initialization of deep networks matters. Generally, a layer's weights are randomly initialized by sampling from a uniform distribution, denoted by $U[-m, m]$ for some $m$ based on the number of inputs, outputs and the activation function. Hence, after minor modifications for different activation functions, we use the following initialization:

$$u_{ik} \sim U\left[-\sqrt{\frac{(1 + \sum_j A_{ji})}{\sum_j (A_{ji} \sum_k A_{jk})}}, \sqrt{\frac{(1 + \sum_j A_{ji})}{\sum_j (A_{ji} \sum_k A_{jk})}}\right], \quad v_{ij} \sim U\left[-\sqrt{\frac{1}{1 + 5c_{in}}}, \sqrt{\frac{1}{1 + 5c_{in}}}\right]$$

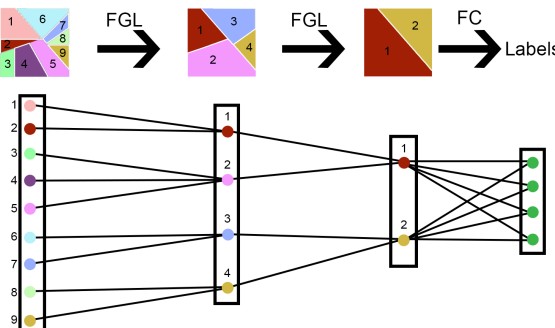

Figure S2: Illustration of FGL. Given the numbered hierarchical segmentation, FGL extracts features for each segment. The inputs are 9 variables corresponding to segments of a square, which are first grouped as $\{\{1,2\}, \{3,4,5\}, \{6,7\}, \{8,9\}\}$. The resulting 4 groups are then grouped into 2 groups using the grouping $\{\{1,2\}, \{3,4\}\}$. The output is passed to a fully connected network to predict labels. Note that intermediate layers can use feature vectors of length greater than 1.

## S5   PARAMETER SHARING

One of the major benefits of using convolution is that it performs parameter sharing - which comes with its own benefits. Adapting FGL to perform parameter sharing is much harder. Typically, a fully connected from $n_{in} \times c_{in}$ numbers to $n_{out} \times c_{out}$ numbers would require $n_{in} \times n_{out} \times c_{in} \times c_{out}$ parameters. But this number is astronomical. To avoid using as many parameters, we decompose the operation into a multiplication by $v$ followed by a Hadamard product with $u$. Doing so reduces the number of parameters to $c_{in} \times c_{out} + n_{in} \times c_{out}$. This is much more tractable, but more reduction might be possible: sharing parameters between groups seems lucrative, unfortunately, different groups can have different sizes and an arbitrary ordering - prevent us from parameter sharing further. If group sizes were constant and an ordering of variables was fixed, it would be possible to further reduce the number of parameters from $\mathcal{O}(n_{in})$ to $\mathcal{O}(groupsize)$.

## S6   CNN ARCHITECTURE

We use a straightforward architecture for convolution - repeated convolutional layers of stride 2 and a kernel size of 4 with appropriate padding, followed by a fully connected network. We found that using `maxpool` for downsampling reduced performance, and so did using non-linear activation functions. A visualization is provided in S1.

