# OpenReview forum: "Towards a Deep Network Architecture for Structured Smoothness"
_ICLR.cc/2020/Conference — Accept (Poster)_

### Official Review · AnonReviewer1 · 2019-10-23
**Official Blind Review #1**

**Rating:** 6

**Review:**

"Towards a Deep Network Architecture for Structured Smoothness" proposes a new layer, dubbed FGL, specifically focused on structured smoothness.
The paper clearly discusses the proposed layer, introduces the necessary formulation, and demonstrates the effectiveness of this layer in a suite of practical applications in fMRI analysis. The writing is clear, the method is well formulated, and performance compared to relevant and recent benchmarks is strong. Thus most of the questions remaining are about the finer details of the proposed layer and method.

What is the sensitivity of FGL to non-optimal group proposals for A? Given the authors reference Aydore et. al., do randomized (or perhaps meta-optimized) groupings perform poorly? Are there any potential ways to jointly-or-iteratively learn the groupings? Is Ward clustering the primary practical method for getting the groupings?

Can the authors discuss a bit the relationship and potential tradeoffs between "early" segmentation architectures (such as FGL), and those often favored in semantic segmentation ("late" segmentation)? Is there a particular reason why "late" segmentation is particularly a poor choice in brain imaging?

My primary concerns come in relation to other methods for structured smoothness such as CRF/MRF, and the trade-offs and drawbacks of imperfect or downright wrong A, and alternative methods for practically finding good A matrices (potentially with an eye to applications outside fMRI). Addressing some or all of these in a section or two of the text body would potentially raise my score.

Some small errors:
Optimg -> Opting Introduction
Guassian -> Gaussian section 3.2

Potential useful references:
CRF as RNN https://arxiv.org/abs/1502.03240
Efficient Piecewise Training https://www.cv-foundation.org/openaccess/content_cvpr_2016/html/Lin_Efficient_Piecewise_Training_CVPR_2016_paper.html
Deep Learning Markov Random Field for Semantic Segmentation https://arxiv.org/abs/1606.07230
Pixel Adaptive Convolutional Neural Networks http://openaccess.thecvf.com/content_CVPR_2019/html/Su_Pixel-Adaptive_Convolutional_Neural_Networks_CVPR_2019_paper.html

**Experience Assessment:**

I have published one or two papers in this area.

**Review Assessment: Checking Correctness Of Derivations And Theory:**

I assessed the sensibility of the derivations and theory.

**Review Assessment: Checking Correctness Of Experiments:**

I assessed the sensibility of the experiments.

**Review Assessment: Thoroughness In Paper Reading:**

I read the paper at least twice and used my best judgement in assessing the paper.

---

> ### Author Response · Authors · 2019-11-15
> **Thank you for your feedback! It is insightful and will help us improve our discussion.**
>
> Relation to MRF’s and CRF’s:
> Probabilistic graphical models are a great approach for capturing known structure in data. We believe our approach is complementary and distinct from the probabilistic approaches. Specifically, the relationship between FGL and MRF/CRF is analogous to the relationship between RNN/LSTM/GRU and HMM’s. Both approaches tackle a similar task. One focuses on a deep neural network for structured representation learning, while the other focuses on direct probabilistic modeling. One advantage of the FGL deep representation learning is fast inference as compared to MRF/CRF, perhaps at the cost of diminished interpretability. On the other hand, probabilistic models clearly capture detailed semantics but can pose the challenge of efficient inference. As mentioned, these methods can be closely related. Constructing a probabilistic method that is equivalent to FGL (such as the RNN<->CRF equivalence you have pointed out) is an interesting question that is left for future work.
>
> Additional discussions on tradeoffs of FGL for structured smoothness (re: early vs. late)
> 1. Computational feasibility: An important point to notice is the dimensionality of the problem. fMRI images are 3D (200k “pixels” in our case), while standard images are 2D - hence, of much lower dimensionality. Thus, to use a neural network (and remain computationally feasible) some form of pooling is required in the forward pass.
>
> 2. Results on downsampled images: As an early experiment, we attempted the same task on 2x downsampled images and saw a ~10% drop in accuracy (for FGL and worse for other models). This implies that there is value in doing this segmentation earlier rather than later.
>
> Effects of selecting A:
> 1. Imperfect A (simulated experiments): To clarify how our results depend on the quality of the A matrix, we constructed an experiment to measure the sensitivity of  FGL is to A in simulated conditions. Here, we generated approximate A by using MiniBatchKMeans using 16, 32 and 48 clusters -- and everything else matching our simulated results (Section 3.2). The resulting accuracy was 61.3%, 66.1%, and 66.3%. Recall, that the ground truth A, with 32 regions/clusters, has 70.2% accuracy. This drop in accuracy still outperforms CNNs and CoordConv (56% accuracy) and FNNs (43% accuracy). The additional drop in accuracy at 16 clusters is expected since we’re trying to represent what happens in 32 regions by just 16 regions. These results have been added to the supplement.
>
> 2. Imperfect A (in fMRI): Indeed, it is true that our method can be affected by the quality of A. We chose ward clustering, itself known to be an approximation, as it is considered one of the best methods for data-driven clustering of fMRI (Thirion et.al., 2014). Our experiments show that it can be very effective. Our simulated results suggest that future improvements to fMRI clustering will most likely serve to improve our results further.
> Taken together, these results suggest that FGL with an inaccurate A will still outperform standard approaches (in all our experiments, both real and simulated)
>
> Your idea of learning a good clustering directly for the supervised task is also an interesting one. Ideally, one can do this using a semi-supervised approach that takes advantage of the wealth of unsupervised data (as is common for fMRI and other biological datasets). We plan to explore this further in future work.
>
> How to find a good A:
> The grouping that should be used in each problem depends on the domain - in the case of fMRI imaging, there is previous work that shows that Ward clustering produces a good grouping, and indeed, it does outperform random groupings (like the ones that Aydore used). Below are some ideas on other domains where we believe FGL can be applied.
>
> 1. Biological images: As mentioned in the manuscript, biological images are often aligned as a standard preprocessing step. Further, anatomical constraints ensure that the coarse structure of the image is similar across samples.  Examples include x-ray images of the lung and CT images of knees, among others. The task of interest is often to automate diagnosis. Extending this work to a broad range of biological problems is an active research interest for us, with very promising preliminary results.
>
> 2. Financial predictions from stock market price data: Stock prices are correlated with each other in interesting ways (across countries, industries, etc). The correlation matrix could be used as A. Reasonable choices include known correlations (e.g., based on country, industry) or estimated correlations (i.e., unsupervised from data).
>
> 3. Face Identification: Consider the problem of face identification with aligned photos (e.g. mugshots). Here, the alignment of the photos also suggests that informative structure, e.g., the location of the forehead, nose, etc., is shared across images, thus associated pixels may be represented as a “group”.

---

### Official Review · AnonReviewer2 · 2019-10-24
**Official Blind Review #2**

**Rating:** 6

**Review:**

This work introduces fixed grouping layers to deep learning models. Unlike convolutional layers, the fixed grouping layer only allows the output to be impacted by the specific inputs associated to it by its group. The paper stays firmly in the area of brain scans with the authors hoping to generalize to other applications later. Their experiments compare to what they consider to be the state-of-the-art approaches to the problem.

I did find Figure S2A helpful to quickly understand some of the simulated data approach, maybe it should be in the paper and not supplementary.

I argue for accepting this paper because the experiments, while focused solely in one discipline, are thorough. I would have liked clearer description of other areas that might benefit in the last sentence.

**Experience Assessment:**

I do not know much about this area.

**Review Assessment: Checking Correctness Of Derivations And Theory:**

N/A

**Review Assessment: Checking Correctness Of Experiments:**

I assessed the sensibility of the experiments.

**Review Assessment: Thoroughness In Paper Reading:**

I made a quick assessment of this paper.

---

> ### Author Response · Authors · 2019-11-15
> **Thank you for your positive feedback!**
>
> On Fig S2A in the main paper: We agree, and decided to put this figure in the supplement due to limited space. We have swapped Fig S2A for Fig 2 in the main paper, as we believe it does a better job of clarifying the main ideas.
>
> On additional potential applications (with details on how one would estimate the grouping matrix) include the following:
> 1. Biological images: As mentioned in the manuscript, biological images are often aligned as a standard preprocessing step. Further, biological constraints ensure that the coarse structure is similar across samples.  Examples include x-ray images of the lung, CT images of knees, among others. The task of interest is often to automate the diagnosis. Extending this work to a broad range of biological problems is an active research interest for us, with very promising preliminary results.
>
> 2. Financial predictions from stock market price data: Stock prices are correlated with each other in interesting ways (across countries, industries, etc). The correlation matrix could be used as A. Reasonable choices include known correlations (e.g., based on country, industry) or estimated correlations (i.e., unsupervised from data).
>
> 3. Face Predictions: Consider the problem of face identification with aligned photos (e.g. mugshots). Here, the alignment of the photos also suggests an informative structure, e.g., the location of the forehead, nose, etc., are generally roughly the same, allowing those pixels to be represented as a “group”.

---

### Decision · Program_Chairs · 2019-12-19

**Decision:**

Accept (Poster)

**Comment:**

The AC has carefully looked at the paper/comments/discussion in order to arrive at this meta-review.

Looking over the paper, the FGL layer is an interesting idea, but its utility is only evaluated in a limited setting (fMRI data), rather that other types of images/data. Also, the approach seems to work on some of the fMRI datasets, on others the performance is on par with the baselines.

Overall, the paper is borderline but the AC believes the paper would be a good contribution to the conference.